# MVDoppler: Unleashing the Power of Multi-View Doppler for MicroMotion-Based Gait Classification

**Soheil Hor, Shubo Yang, Jaeho Choi, and Amin Arbabian**
Department of Electrical Engineering
Stanford University
Stanford, CA 94305
`(soheilh,shuboy,jhochoi,arbabian)@stanford.edu`
Project page: `https://mvdoppler.github.io`
Code and dataset toolbox: `https://github.com/soheilhr/MVDoppler`

## Abstract

Modern perception systems rely heavily on high-resolution cameras, LiDARs, and advanced deep neural networks, enabling exceptional performance across various applications. However, these optical systems predominantly depend on geometric features and shapes of objects, which can be challenging to capture in long-range perception applications. To overcome this limitation, alternative approaches such as Doppler-based perception using high-resolution radars have been proposed. Doppler-based systems are capable of measuring micro-motions of targets remotely and with very high precision. When compared to geometric features, the resolution of micro-motion features exhibits significantly greater resilience to the influence of distance. However, the true potential of Doppler-based perception has yet to be fully realized due to several factors. These include the unintuitive nature of Doppler signals, the limited availability of public Doppler datasets, and the current datasets' inability to capture the specific co-factors that are unique to Doppler-based perception, such as the effect of the radar's observation angle and the target's motion trajectory. This paper introduces a new large multi-view Doppler dataset together with baseline perception models for micro-motion-based gait analysis and classification. The dataset captures the impact of the subject's walking trajectory and radar's observation angle on the classification performance. Additionally, baseline multi-view data fusion techniques are provided to mitigate these effects. This work demonstrates that sub-second micro-motion snapshots can be sufficient for reliable detection of hand movement patterns and even changes in a pedestrian's walking behavior when distracted by their phone. Overall, this research not only showcases the potential of Doppler-based perception, but also offers valuable solutions to tackle its fundamental challenges.

## 1 Introduction

Today's perception systems typically benefit from the integration of high-resolution optical sensors and advanced deep neural networks surpassing even human perception in various applications. However, such systems also suffer from inherent limitations of optical system design and the physics of energy propagation in 3D space, which cannot be solved even using the most advanced perception neural networks. As an example, camera-based perception systems typically rely on extracting geometrical (spatial) features of the environment with the effective observed resolution decreasing quadratically as a function of distance. This results in a significantly lower performance in long-range perception applications. Alternatively, there exist perception approaches that do not rely on geometrical features and therefore offer enhanced resilience against distance-related information loss.

37th Conference on Neural Information Processing Systems (NeurIPS 2023) Track on Datasets and Benchmarks.

One such approach is Doppler-based perception, enabled by the new generation of commercially available, high-resolution, mm-Wave coherent radar systems. Unlike camera-based systems that primarily capture an object's shape, location, and color information (such as the shape and pose of a human body, the color of a car, or the dimensions of a helicopter), Doppler-based systems detect the motions exhibited by subjects and their components (such as the motion of human hands or legs during walking, the movements of a car's wheels, or the rotation of a helicopter's rotors).

One application of micro-Doppler analysis methods that has been less investigated is using automotive radar sensors, already present in cars, for predictive accident prevention. Such approaches involve extracting high-level information that can potentially be used to predict pedestrians' traffic behaviour and their interactions with other pedestrians and cars on the road. Examples include, analyzing a pedestrians' hand or leg micro-movement patterns to determine whether the pedestrian is accompanied by a child [1] or, as investigated in this paper for the first time, detecting pedestrians distracted while texting on their phones. While Doppler-based perception is relatively new compared to vision-based systems, it offers a fundamentally novel approach to perception and activity recognition, particularly in long-range applications.

Although there are several advantages to Doppler-based perception, it also comes with its set of unique challenges, which have limited its real-world applications. One challenge is that Doppler measurements are "directional", meaning that the micro-motion information captured by a Doppler sensor depends on the radar's observation angle (a function of subject's relative location to the radar) and the relative direction of the movement of the subject. This challenge is well-known in the radar community, but as it will be shown in the next section, is rarely addressed or mitigated in the literature. Another significant challenge in Doppler-based perception is that micro-motion data is not human-readable. This is in contrast to optical sensors that generate 2D or 3D geometrical (spatial) intuitive representations. This has made it impractical to design hand-crafted micro-motion features, especially for more complex perception tasks. Deep learning methods have addressed this issue by eliminating the need for hand-engineered features, but they require access to large and representative datasets. Publicly available Doppler datasets today are typically very limited in size, and more importantly, lack diversity in capturing co-factors unique to Doppler-based perception, such as the effect of the relative location and trajectory of subjects discussed earlier.

In this paper, we release one of the largest public multi-view micro-Doppler datasets to date, along with single-radar and multi-view baseline perception models. Our work showcases the potential of off-the-shelf mm-Wave radars for complex Doppler-based perception tasks. In specific, this paper's contributions are summarized as follows:

**The first multi-view Doppler dataset for micro-motion-based gait classification**
The published dataset is one of the largest public micro-Doppler datasets in context of human gait analysis and the only multi-view micro-Doppler dataset publicly available in this context to date.

**Baseline neural networks for real-time hand movement and distracted pedestrian detection**
We introduce baseline neural networks for a standard task of hand movement detection, and a task never explored before: distracted pedestrian detection. The baseline models provided in this paper can make predictions in less than a second. This highlights the potential of such methods for virtually real-time Doppler analysis applications.

**A new benchmark for the effect of location and walking trajectories on gait Doppler signatures**
The published dataset is unparalleled in capturing the effect of different relative locations and walking directions on the walking Doppler signature of subjects. Combining the multi-view aspect of the published dataset and its comprehensive coverage of different locations and walking trajectories, we also provide the first *trajectory-agnostic* micro-Doppler-based classifier in the context of gait micro-Doppler analysis.

## 2 Relevant Work

### 2.1 Gait micro-Doppler Datasets

The overview of relevant gait micro-Doppler datasets is shown in Table 1. One main factor separating different datasets in the context of gait classification is the definition of classes. In most Doppler datasets today, the gait classes are limited to simple actions (e.g., fast vs slow walking, or crawling vs jumping) which can be easily classified using the *macro-motions* of the body (mainly torso). In

contrast, hand micro-motion detection is a much harder task, especially in presence of other sources of micro-motions, like movement of the legs while walking. However, most existing works focus on detecting hand while standing [2][3][4]. Our work is one of the few public datasets that include the non-trivial task of hand movement detection for a walking subject, and the first dataset to investigate changes in the walking pattern of pedestrians while distracted by their phones.

Another factor highlighting this proposed dataset is the classification latency of the baseline methods. While most papers aim to classify motion snapshots of around three seconds [5] [2] [4], we aim to achieve real-time Doppler-based perception by reducing the size of the micro-motion snapshots to less than one human walking cycle (500-1000ms).

Moreover, trajectory and location coverage are often overlooked in existing datasets while Doppler measurements are very sensitive on radar's observation angle. There are three critical factors to consider in this context: range coverage, azimuth coverage, and the coverage of participants motion trajectories. Nevertheless, none of the existing datasets have considered these three aspects simultaneously. Vishwakarma et al. [2] select a Region of Interest (RoI) spanning 1-10m in range, but instead of complete coverage of different ranges, they measure random walking patterns of the participants within the RoI. Chakraborty et al. [4] proposed the dataset containing activities with discrete operation ranges and motion directions (i.e., $0°$, $±15°$, $±30°$, $±45°$) from the radar. The dataset presented in this paper enjoys a comprehensive coverage of all of these factors, providing a new benchmark for examining the effect of location and walking trajectories on gait Doppler signatures.

Table 1: Comparison of public gait micro-Doppler datasets.

| Paper | Hand Micro-motion | Classification Snapshot length | Classification Latency | Location and Trajectory Coverage | | | Participants Number | Dataset Size |
|---|---|---|---|---|---|---|---|---|
| | | | | Across Range | Across Azimuth | Motion Direction | | |
| Gurbuz et al. [5] | - | 3s | Fast | ✓ | - | - | S | M |
| Vishwakarma et al. [2] | WS | 2.7s | Fast | ✓ | - | R | M | S |
| Yang et al. [3] | WS | - | - | - | - | - | L | M |
| Gambi et al. [6] | WW | 16s | Slow | ✓ | - | - | L | S |
| Bhavanas et al. [7] | - | 3.7s | Fast | R | R | - | L | L |
| Chakraborty et al. [4] | WS | 3s | Fast | R | - | ✓ | L | M |
| **MVDoppler (Ours)** | WW | 0.64s | Realtime [2] | ✓ | ✓ | ✓ | M | L [1] |

| | | | |
|---|---|---|---|
| | WS: While Standing | WW: While Walking | -: not specified |
| | Slow: >5s | Fast: 1-5s | Realtime: <1s |
| | ✓: Full coverage | R: Random | |
| | S: <10 people or <10,000s | M: 10-20 people or 10,000-20,000s | L: >20 people or >20,000s |

## 2.2 Multi-Radar Fusion

As mentioned earlier, Doppler-based perception using radar is dependent on the subject's relative location and movement direction. One approach to deal with this issue is to fuse information from multiple radars. Multi-radar fusion techniques can typically be categorized into two approaches: fusion of co-located perpendicular radars for improved angular resolution in both elevation and azimuth dimensions [8, 9], and fusion of spatially distributed radars with the goal of achieving better coverage over the RoI [10, 11, 12, 13, 14]. Our work falls under the later group of spatially distributed radars. Early approaches demonstrated the efficacy of fusion from distributed radars at the simulation level, particularly in terms of through-the-wall sensing [10, 11] and ghost mitigation [12]. More recently, the focus has been shifted towards real-world multi-radar setups, with [13] integrating signals from two off-the-shelf radars based on graph matching techniques, while [15] leveraged ensemble learning for improved activity perception within a small RoI. Building upon prior work, this study proposes a multi-view approach to Doppler-based gait analysis. We have specifically designed a multi-radar setup that not only eliminates the blind spots of each radar but also ensures

---

[1]MVDoppler includes over 10.5 hours (37,800 seconds) of simultaneous data capture from two radars with complementary views resulting in over 54,000 multi-view snapshots. For single-radar classification tasks, this translates to an equivalent of 21 radar-hours or over 108,000 single-radar recorded snapshots.

[2]Latency in the context of gait analysis is typically determined by the minimum observation window size required by the perception model to make an accurate classification.

that the information from both radars is complementary with the goal of achieving a truly *trajectory and location-agnostic* sensing system. To the best of our knowledge, this goal has not been achieved by any single-radar or multi-radar system to date.

## 3   Dataset

### 3.1   Classes

To showcase the power of Doppler-based perception for real-time analysis of complex pedestrian gait patterns, we defined four classes and two tasks as shown in Figure 1(a). A total of 13 volunteers were recruited, All providing informed consent per IRB guidelines and agreeing to have their data included in a publicly available dataset[3]. The four classes defined in this study include normal walking ("Normal"), phone call ("Phone Call"), hands in pockets ("Pockets"), and texting ("Texting"). Each class represents a common activity undertaken by pedestrians in a typical traffic scenario. Normal class is intended to capture the natural walking patterns of subjects, allowing for complete freedom of movement in their hands and legs. The Phone Call class is similar to Normal, except for the assumption that the participants are holding their phone to their ear with one hand, leaving only one hand free. Pockets class assumes that the participants' hands are in their pockets, restricting hand movement entirely. Lastly, Texting class assumes the participants are using both hands to text on their phones while walking. To better capture the effects of distracted walking on gait patterns, we asked participants to keep their eyes fixed on their phones rather than maintaining their typical attentive walking behavior.

We defined two classification tasks based on the four classes. The first task is hand movement detection ("Hand"), which aims to distinguish classes with no hand movement (Pockets and Texting) from classes with one or more moving hands (Phone Call and Normal). The second classification task ("Distract"), which aims to detect changes in the walking patterns of individuals when they are distracted by their phones. In absence of hand-movement signatures, we expect the perception models to pick up more subtle cues in the distracted pedestrian's walking pattern, making this task significantly harder comparing to the "Hand" task.

### 3.2   Data Capture Setup

Figure 1(c) illustrates the multi-radar setup and its positioning with respect to the RoI. For our experiments, we used two off-the-shelf FMCW radars (AWR1843 from Texas Instruments), and a HD stereo camera (ZED from Stereolabs). The two radars were designed to operate at two different frequencies (77-78GHz for radar0 and 79-80GHz for radar1), to ensure the simultaneous operation of both radars at a high chirping duty-cycle (>95%) without any risk of interference.

As it will be shown, symmetrical coverage of the RoI by both radars is critical in this study. This is achieved by selecting a rectangular area with a 10m extent in both range and azimuth of each radar, and utilizing the entire width of each radar's Field of View (FoV) in azimuth ($\pm 45°$). Assuming the average height of a human to be less than 2m, both radars are set at a height of 1m to ensure the Doppler signatures of both hands and legs can be properly captured. Considering the FoV of each radar in elevation ($\pm 15°$), we chose the minimum distance of the RoI to each radar to be at 5m. This guarantees the full utilization of the vertical beam width of each radar while eliminating the effect of the beam shape on received signals in close ranges [3].

### 3.3   Experiment Design

To cover all possible walking directions and trajectories within the RoI, we defined seven walking patterns shown in Figure 1(b). Participants were instructed to walk at their natural speed and cover the entire RoI while repeating each walking pattern for all activity classes. Note that these patterns were carefully selected to be complementary from the perspective of each radar, ensuring a natural balance between the distribution of data across radars and walking patterns.

---

[3]More information in the supplementary materials

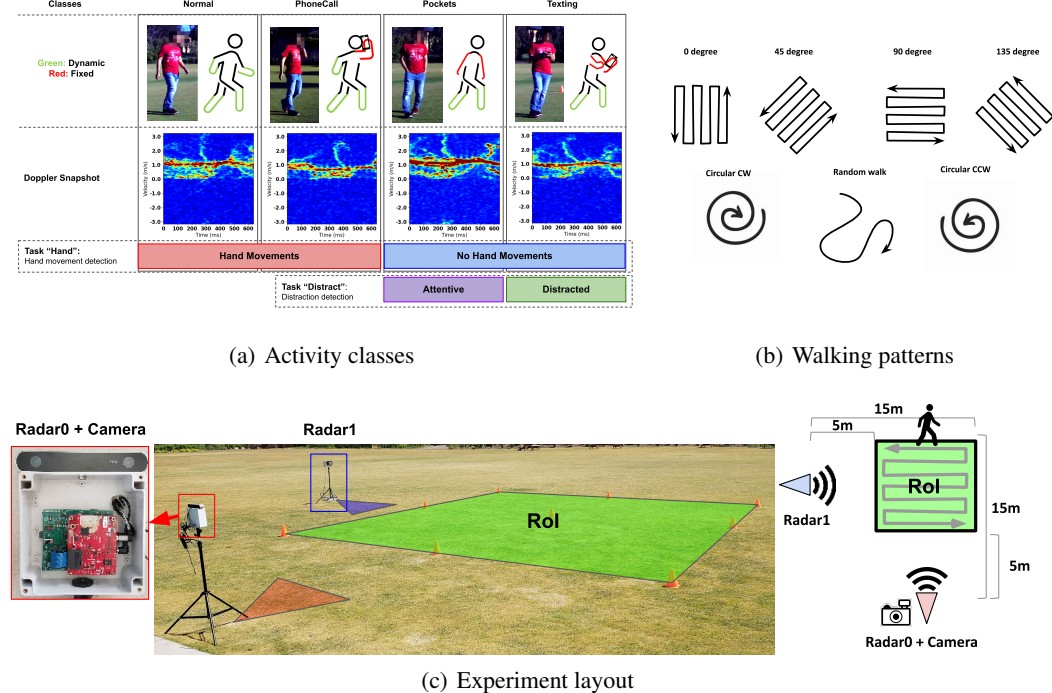

(a) Activity classes           (b) Walking patterns

(c) Experiment layout

Figure 1: Experiment Design. (a) The activity classes and corresponding classification tasks. High-resolution Doppler snapshots are not human-readable, but can be sufficient for real-time motion-based activity classifications. (b) The selected walking patterns. (c) RoI and its position relative to the multi-radar setup.

## 3.4 Dataset Statistics

In Figure 2(a), we present a histogram showing the number of snapshots captured at each location within the RoI. With a uniform coverage of samples and over 90 snapshots at each location bin, the dataset provides sample size to accurately measure the effect of relative location on classification results.

Figure 2(b) illustrates the distribution of snapshots captured for different walking velocities. It can be observed that the dataset exhibits a symmetrical coverage within average walking speeds of our subjects ($1 \pm 0.25m/s$ indicated by the white dashed lines) with over 90 snapshots per bin for the primary linear walking directions ($0°, 45°, 90°$, and $135°$, and their counterparts). This is evident that the dataset provides sufficient sample-size to accurately measure the effect of walking trajectories on the gait classification performance.

Figure 2(c) shows the distribution of average walking velocity over the RoI. The velocity is mostly uniform within the RoI, with the exception of low-speed regions around the edges of the RoI where participants stop and turn to avoid exiting the field. Minimizing the correlations between location and walking velocities of subjects in the training data is a critical component in ensuring generalization power of Doppler-based classifiers, a factor commonly overlooked in the capture and analysis of micro-Doppler datasets.

## 4 Single-Radar Baseline Neural Networks

### 4.1 Pre-Processing

The initial step involves extracting the micro-Doppler signatures from each radar's received raw signal. We employ the CFAR algorithm [16] to detect the participant locations, and then use short-time Fourier transform on the slow-time axis to extract the gait micro-Doppler spectrograms. Then, the spectrogram's magnitude is log-transformed and normalized as a grayscale input to deep neural networks[3].

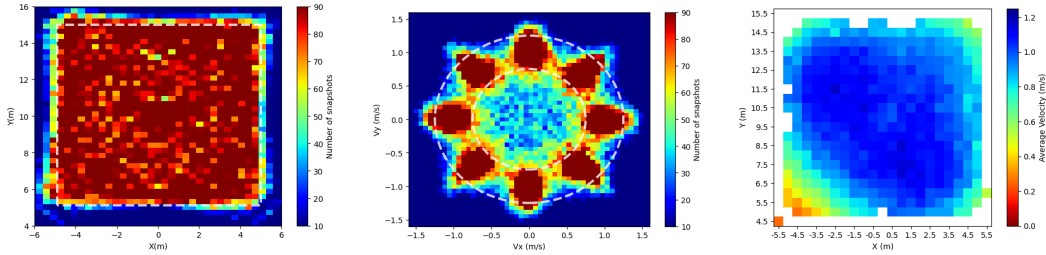

(a) Number of snapshots captured at each location.

(b) Number of snapshots captured for different motion directions and velocities.

(c) Average walking velocity over the RoI.

Figure 2: Dataset statistics. The captured dataset provides a comprehensive coverage across all walking directions and locations within the RoI.

The dataset is collected in the form of 10-second walking episodes, which are further divided into 1.28-second clips with a 50% overlap. To avoid the classifiers being sensitive to the relative start phase in walking cycles, we augment each training batch by randomly cropping each clip into a snapshot of 640ms. Our experiments show that this simple augmentation technique is highly effective in enhancing the generalization of Doppler-based gait classifiers on unseen data.

## 4.2 Gait Classification

Figure 1(a) displays examples of micro-Doppler snapshots captured for the four classes. These snapshots cannot easily be distinguished by untrained human eyes, but contain high-dimensional micro-motion signatures of human gait patterns that can be leveraged by Deep neural networks for complex classification tasks. Hence, we utilize MobileNetV2 [17] as the backbone model for both Hand and Distract tasks. We report the 4-fold cross-validation accuracy across different subjects in Table 2. Remarkably, the classifier achieves reliable hand-movement detection accuracy of 85.25% across subjects by analyzing Doppler snapshots in a window as short as a single walking step. Moreover, Figure 3 demonstrates that using longer windows for inference increases the accuracy to 97.9%±3.07% (accuracy ± standard deviation).

The performance for the more challenging Distract task is naturally lower, yet still statistically significant across subjects (accuracy of 62.55%±5.0% across two radars). This task aims to demonstrate the feasibility of using micro-motion signatures to capture effects beyond simple hand or leg movements, and pose a challenge to encourage advanced micro-Doppler analysis techniques. As shown in Figure 3, similar to the Hand task, increasing the observation window length leads to an accuracy of 72.05%±10.05%.

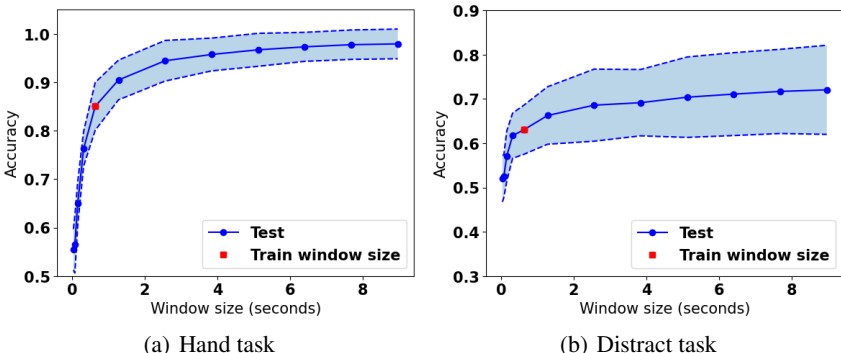

(a) Hand task

(b) Distract task

Figure 3: Hand and Distract tasks results across different inference window sizes. The baseline neural networks trained for real-time operation can also be used for slower but more accurate classification.

Table 2: Classification accuracies (accuracy % ± standard deviation).

| | Single-Radar | | Multi-View Radar Fusion | | | | |
|---|---|---|---|---|---|---|---|
| Tasks | Radar 0 | Radar 1 | Input-level - | Late-level Average | Multi-level Average | Late-level Transformer | Multi-level Transformer |
| Hand | 86.0±4.6 | 85.7±4.3 | 85.2±5.1 | 87.7±4.8 | 88.5±4.5 | 88.6± 4.8 | **90.2**±4.6 |
| Distract | 63.5±6.0 | 61.5±4.9 | 60.0±6.9 | 65.8±7.5 | 66.9 ±6.4 | **67.4**±5.6 | 66.9±6.3 |

## 5 Multi-Radar Baseline Neural Networks

### 5.1 Multi-Radar Fusion Methodology

In this section, we provide baseline models for the semantic fusion of multi-view Doppler signals, to address their inherent dependency on subject's location and trajectory.

Let $f(\cdot\,;\boldsymbol{\theta})$ denote a baseline embedding network with its internal parameters $\boldsymbol{\theta}$, designed to project an arbitrary micro-Doppler snapshot $X \in \mathbb{R}^{T \times F}$ into the latent space $f(X\,;\boldsymbol{\theta}) = \{\eta_1, \eta_2, \ldots, \eta_L\}$, where $\eta_l$ is the representation from the $l$-th encoding layer. The multi-view representations $f(X^{R1}\,;\boldsymbol{\theta}) = \{\eta_1^{R1}, \eta_2^{R1}, \ldots, \eta_L^{R1}\}$ and $f(X^{R2}\,;\boldsymbol{\theta}) = \{\eta_1^{R2}, \eta_2^{R2}, \ldots, \eta_L^{R2}\}$, obtained from the multi-view micro-Doppler modalities $X^{R1}$ and $X^{R2}$, respectively, can be integrated across a range of different fusion levels and methods.

We demonstrate multi-view fusion across three different embedding levels: 1) input-level fusion, where the inputs $X^{R1}$ and $X^{R2}$ are channel-wise concatenated and embedded via the same single-view baseline network, 2) late-level fusion, where the features are combined solely under the last layer $L$, and 3) multi-level fusion, contemplates feature embeddings across all levels wherever the features are resized, e.g., every representation level right after pooling.

One approach for multi-view fusion is to simply combine the corresponding representations from the two radars by summing them together, i.e., $(\eta_l^{R1} + \eta_l^{R2})/2$. However, given that the radar signals captured from different spatial nodes pose inequivalent significance with respect to the heading direction of a target, it may be more reasonable to assign dynamic weights across the multi-view sequences. This intuition leads us to introduce a transformer-based fusion pipeline that merges the tokenized form of $\eta_l^{R1}$ and $\eta_l^{R2}$ via token-wise concatenation [18], followed by self-attention layers [19] to achieve cross-attention between the multi-view inputs as well as context-aware temporal attention.

The combinations across different fusion levels and methods yield five multi-view radar fusion variants: input-level fusion, late-level average-based fusion, multi-level average-based fusion, late-level transformer-based fusion, and multi-level transformer-based fusion.

### 5.2 Experimental Analysis

Table 2 summarizes the classification performances for each fusion method. The results demonstrate that the multi-view setting has holistic performance improvements in comparison with a single-radar system. Of particular note is the trend that the use of adaptive fusion strategies at multiple-levels leads to greater performance gains. Comparing with single-radar baselines, the proposed multi-level transformer-based fusion baseline yields in significant improvements of around 5% for both for Hand and Distract tasks.

## 6 Results and Discussion

It is worth noting that the accuracies shown in Table 2 represent the average classification performance across the entire RoI and all velocities and directions. In this section, we provide an in-depth analysis of single-view and multi-view classification results (for the best models presented in the previous section) with respect to the effect of location and velocity. Figure 4 illustrates that the single-radar classification accuracy decreases towards the edges of the RoI for both radars. This accuracy loss is partly due to the reduced velocities at the edges of RoI, and in part due to the decreased signal-to-noise ratio (SNR), which can be mitigated using beamforming techniques or increasing radar's transmit power. Figure 5 illustrates the accuracy distribution of the Hand task across the two-dimensional

walking speed and direction of the subjects. The results highlight the significant sensitivity of single-radar classification to such factors. The accuracy substantially drops for subjects with tangential directions from the radar, owing to Doppler's radial sensing nature. This effect is also clear in the accuracy vs walking angle plot in Figure 6.

In contrast to single-radar baselines, Figure 4(c) and Figure 5(c) show that multi-view sensing has great robustness at the edges of the RoI as well as on the tangential velocities. This is evidence that multi-view radar sensing techniques can compensate for the limitations of single-radar approach, and eliminate single-radar location and trajectory dependence. This result is further elucidated in Figure 6, representing the average accuracy of the Hand and Distract tasks as a function of walking direction relative to Radar0. Notably, the two radars exhibit complete complementary behavior specially for the blind spots of each radar (i.e., $\pm 90°$ for Radar0 and $0°$ and $180°$ for Radar1). The multi-radar fusion consistently achieves higher accuracy than the best performance of either sensor alone, irrespective of the directions.

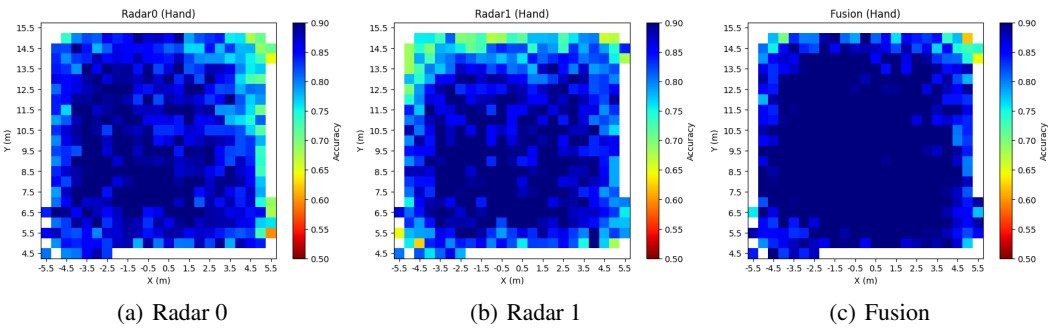

(a) Radar 0          (b) Radar 1          (c) Fusion

Figure 4: Classification accuracy across the RoI for the Hand task. The classification performance of the multi-view fusion approach is uniform across the RoI.

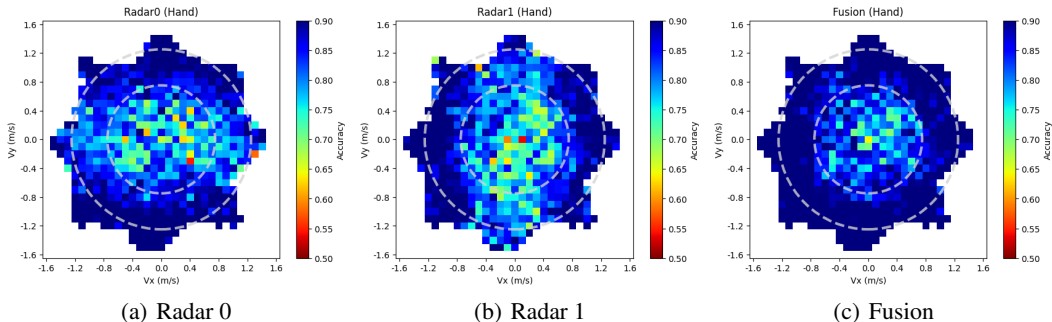

(a) Radar 0          (b) Radar 1          (c) Fusion

Figure 5: Hand task classification accuracy across walking velocity magnitudes and directions. Single-radar classification performance is sensitive to the subjects direction of motion while the multi-view approach is independent of the subjects moving direction.

## 7 Limitations

In this study, we focused on exploring the influence of subject's location and trajectory co-factors on gait analysis. However, there exist several other co-factors that we did not investigate. For instance, we did not examine the impact of multi-path and clutter in different environments or the effect of movements of the radar itself on the Doppler signature.

Another limitation of the presented dataset is related to the sample size and diversity of participants. Although the presented results demonstrate statistical significance as indicated by the confidence intervals in Table 2, the dataset could enjoy a larger and more diverse participants population.

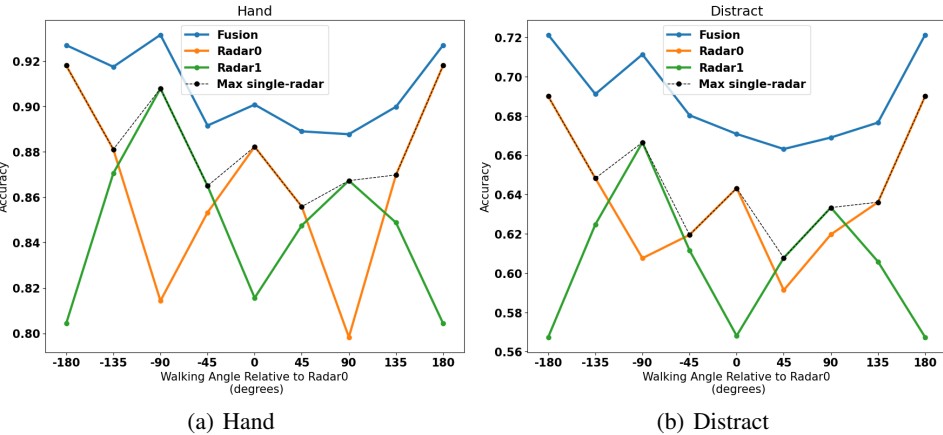

|  (a) Hand | (b) Distract |

Figure 6: Accuracy of Hand and Distract tasks across walking angles. Radars complement each other in their sensitivity to effect of different walking directions. Fusion-based approach outperforms both single radars across all walking directions.

Moreover, it is important for readers to note that the classes defined in this paper do not fully encompass the breadth of diversity and complexity inherent in the two tasks. The selected classes and the tasks introduced in this study were specifically designed to encompass varying levels of difficulty in Doppler-based pedestrian gait analysis. They serve as an initial step towards utilizing Doppler-based methods for advanced perception tasks.

# 8 Conclusion

This study introduces a novel micro-Doppler dataset, consisting of multiple views, for Doppler-based pedestrian gait classification. Our research focuses on two challenging tasks: real-time hand movement detection and distracted pedestrian detection. We establish neural network baselines for these tasks and investigate the impact of Doppler directionality on single radar classification performance—a factor often overlooked in existing Doppler-based classification literature.

To address the challenges posed by relative location and velocity, we propose multi-view neural network baselines based on fusion transformers. Through this approach, we achieve trajectory-agnostic, real-time pedestrian gait classification. The results we present demonstrate generalization across different subjects and various walking trajectories, as evidenced by the narrow confidence intervals of 5-7% reported for both tasks.

The availability of our dataset, along with the corresponding baseline neural networks, lays a solid foundation for future research in real-time Doppler-based perception. This research avenue offers an alternative to conventional vision-based perception approaches. Overall, our work contributes significantly to the field of Doppler-based pedestrian gait classification and opens up opportunities for further advancements in real-time Doppler-based perception.

## Acknowledgments and Disclosure of Funding

This work has been supported by Samsung Electronics (Samsung Semiconductor USA).

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
