# 1 Supplementary Material

## 1.1 Social Impact

In this paper we demonstrated the potential of motion-based perception as an alternative to geometrical (spatial) perception approaches. Motion-based perception can be used in a diverse set of robotics and remote sensing applications. Moreover, it has the unique advantage of being resilient to effect of distance and environmental factors (e.g. weather and lighting conditions). The methods proposed are specially useful for long-range perception applications like autonomous driving, perimeter security, or privacy preserving activity monitoring in public or private spaces. In specific, our focus was on predictive accident prevention applications for autonomous driving with the goal of making the roads safer for pedestrians.

Like any other remote perception technology, there are also risks involved with misuse of radar-based perception especially in the context of activity monitoring. Radars in principle are capable of analyzing activities of subjects from behind walls, from long distances and around corners. Therefore they potentially can be used for malicious applications like surveillance and activity monitoring of humans without their knowledge or consent. This motivates research on methods for detecting and blocking radar signals in private spaces.

## 1.2 Ethics Statement

This study involved recording radar data from adult volunteers performing typical pedestrian activities in a public space. We did not see risk of any significant harm to the participants or any privacy concerns for each participant. Nevertheless, we acquired approvals from Stanford University's IRB and privacy offices before capturing any data (IRB protocol number 64473). Each participant signed a consent form agreeing to have their recordings published as part of a public dataset before taking part in the study. Doppler signals published as part of this dataset are categorized as unidentifiable sensing modalities and any attempt to re-identify the subjects is not intended by the authors.

## 1.3 URL to Website

Please see project page: `https://mvdoppler.github.io` to access the data and the corresponding Github repo `https://github.com/soheilhr/MVDoppler` for the code and dataset toolbox.

## 1.4 Statement of Responsibility

The authors assume responsibility outlined by the IRB and Privacy guidelines of Stanford University regarding capture and publishing this dataset. The dataset is published under CC BY-NC-ND license. The code is published under Apache License 2.0.

## 1.5 Data Maintenance Plan

The dataset is hosted on a Google Drive space maintained by Stanford University. In case of data relocation, authors will ensure that a link to the dataset's new location is present in the project page mentioned above. Users can submit issues with the dataset both by submitting a git issue ticket in the corresponding Github repository and by emailing the authors.

## 1.6 Data Format and Instructions on How to Access and Read Data

Doppler snapshots are stored in hdf5 format. Labels and all other metadata are saved in .csv and .json formats. Detailed instructions on how to download and read the dataset together with example codes are available at: `https://github.com/soheilhr/MVDoppler`

## 1.7 Implementation Details

### 1.7.1 Dataset Cleanup and Statistics Extraction

As it was explained above, MVDoppler is captured in 10 second episodes which are then cut into smaller 1.28 second spectrograms each including one cycle of the human gait. We performed data

quality control and cleanup on 3 different scales: During the data capture we periodically sanity checked the captured data to ensure that the size of the captured data matches our expectation. We then eliminated episodes which were significantly shorter than the expected length (due to packet-loss in data transmission, radar malfunction or other issues in saving the data). We then cut the cleaned-up episodes into smaller snapshots. Each snapshot was also manually checked one more time to ensure it contains signal from the subjects. During the process above, we also extract location and velocity point-clouds for each snapshot. The point clouds are then used to fuse data across the two radars and estimate the 2D location and velocity vectors for multi-view snapshots. 2D location and velocity vectors together with observed average SNR per view are also included as part of the dataset in the form of a curated design table.

### 1.7.2 Training and Validation Details

In order to ensure unbiased training of baseline neural networks on the presented dataset, we randomly split the dataset into 4 folds across subjects. We then randomly choose 10% of episodes of the training folds to be hold-out as the validation set.

To help with faster convergence, we trained the models starting with a pre-trained MobileNetV2 (with imagenet weights). We then normalized each snapshot and duplicated it across the channel dimension simulating the RGB channels of the networks inputs. Finally, we augment the inputs for relative walking phase in the gait cycle by randomly cropping each 1.28second radar spectrogram into a snapshot with length of 640ms.

For all single-radar models we used the Adam optimiser together with cosine annealing learning rate decay and cross-entropy loss. We took a multi-step approach to train the baseline models: We first tuned each hyperparameter shown in Table 1 using the validation performance of single-radar models based on fold0. We then trained a model for each fold using an early stopping method with patience of 10 epochs and a delta of 0.01 and based on the corresponding fold's validation loss.

We used the pre-trained single-radar neural networks for fusion experiments. Based on the multi-view representations embedded in weight-shared networks, we applied five fusion baselines. For multi-level fusion, we adopted $1 \times 1$ convolution to ensure that the output of each embedding level corresponded to that of the last layer. The model was optimized with the same training procedures as those employed in the single-radar case, with only the exception of batch size which was set to 32. On average each training session of the single radar required around 1.7 hours of GPU time on an RTX2080ti GPU. The multi-radar experiments required varying durations with respect to the fusion strategies, with the fusion methods of input, late-average, multi-average, late-transformer, and multi-transformer taking 0.9, 0.8, 1.0, 1.4, and 2.0 hours on an RTX3090 GPU, respectively.

### 1.8 Relationship Between Observed Doppler and The Relative Direction of Motion

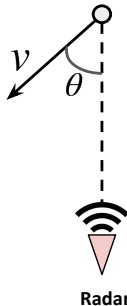

Figure 1: Target's motion relative to the Radar's line-of-sight axis.

The Doppler effect, also known as Doppler shift, refers to the alteration in frequency experienced by a wave, whether it's light, sound, or electromagnetic (EM) waves, when the source is in relative motion with respect to the observer. For instance, if the source moves towards an observer at rest, successive wave crests will reach the observer in shorter intervals than their predecessors, causing the observer to receive waves of a higher frequency than those emitted by the source. As a result, the

Doppler frequency in EM waves is contingent upon the relative motion between the source and the observer. As shown in Figure 1, the Doppler frequency shift $f_D$ in radars can be described as

$$f_D = -\frac{2v\cos(\theta)}{c},$$ (1)

in which $v$ is the absolute velocity of the object, $\theta$ is the angle between the object and line-of-sight (LoS) axis of the radar, and $c$ is speed of light. It is worth mentioning that the Doppler shift is dependent both on the object's absolute velocity $v$ and its relative angle from the radar $\theta$. Thus, when a person is walking directly towards the radar, $f_D$ is maximized, while walking perpendicularly with radar LoS, results in $f_D$ approaching 0.

## 1.9 Experiment and Setup Parameter Design

Table 1 provides details on the parameters selected for our experiments. The designed waveform controls the resolution of each radar in range, angle, and Doppler. Since in our experiments simultaneous operation of two radars was required, we dedicated only 1GHz of bandwidth to each radar instead of utilizing the entire supported bandwidth. We then designed other parameters with the goal of maximizing the Doppler resolution, maximizing the frame duty cycle, and reducing the frame rate.

One important note in the design of the proposed experimental setup is that because of the different operating frequencies of the two radars, the antenna beam patterns of the two radars do not align in the elevation axis. To mitigate this issue, we tilted radar1 downwards for around 10 degrees, this resulted in reducing the average SNR mismatch between the two radars to less than 1dB.

Table 1: Selected Parameters

| Waveform Parameters | | Pre-processing Parameters | |
| --- | --- | --- | --- |
| Operating frequency | Radar0: 77.1-78.1 GHz Radar1: 79.4-80.4 GHz | Window size | 128 chirps (slow-time) 1.5 m (range) 640 ms (time) |
| Chirp bandwidth | 1 GHz | Spectrogram FFT size | 128 |
| Samples per chirp | 128 | Spectrogram window overlap | 87.5% |
| Chirps per frame | 128 | Spectrogram windowing function | Hamming |
| Chirp rate | 3.344 kHz | Neural Networks Training Parameters | |
| Frame rate | 25 fps | Learning rate | 1.00E-04 |
| Antennas | Tx: 1 Rx: 1 | Epochs | 50 |
| Scene Parameters | | Batch size | 64 (single-radar) 32 (multi-radar) |
| Radar locations | Radar0: (5,0) m Radar1: (0,10) m | Optimiser | Adam with Cosine LR Annealing |
| Radar height | 1m | $\epsilon$ | 1.00E-08 |
| Radar vertical tilt angles | Radar0: 0 degrees Radar1: -10 degrees | $\beta1$ | 0.9 |
| Camera location | Co-located with Radar0 | $\beta2$ | 0.999 |

## 1.10 Participant Statistics

Table 2 provides detailed information regarding the participant diversity and number of snapshots available per participant. As it can be seen, although the participants were not instructed to walk

with any specific speed, the cross-subject variance of the natural walking speed is relatively low comparing to the variance of each subject's walking speed across different trajectories. The same observation holds for the cross-subject variance of average SNR comparing to the SNR variance of subjects across different trajectories. This is an evidence for importance of capturing the effect of location and trajectories in Doppler datasets.

Table 2: Participant Statistics

| Subject | Sex | Age | Height (cm) | BMI | Number of Snapshots | | | | SNR (dB) | Velocity (m/s) |
| | | | | | Normal | Phone Call | Pockets | Texting | | |
|---|---|---|---|---|---|---|---|---|---|---|
| 0 | male | 26 | 175 | 23.6 | 1508 | 1560 | 1480 | 1556 | 40.69±4.14 | 0.92±0.23 |
| 1 | male | 29 | 182 | N/A | 2028 | 2338 | 2340 | 2278 | 40.33±4.94 | 1.07±0.30 |
| 2 | male | 34 | 180 | N/A | 2080 | 2204 | 2338 | 2286 | 40.70±4.18 | 0.95±0.29 |
| 3 | male | 30 | 165 | 22.3 | 2184 | 2340 | 1586 | 2310 | 40.87±4.52 | 0.85±0.17 |
| 4 | female | 21 | 165 | N/A | 2596 | 2284 | 2328 | 2326 | 40.24±4.58 | 0.85±0.26 |
| 5 | female | 25 | 167 | 21.2 | 4102 | 2312 | 2340 | 2286 | 39.14±4.62 | 1.11±0.31 |
| 6 | male | 29 | 170 | 25.1 | 1476 | 1448 | 1448 | 1414 | 40.72±4.69 | 1.03±0.21 |
| 7 | male | 24 | 172 | 16.9 | 2314 | 1524 | 1554 | 2300 | 39.34±4.48 | 0.90±0.23 |
| 8 | male | 27 | 180 | N/A | 1508 | 1534 | 1560 | 1530 | 41.16±4.74 | 1.05±0.24 |
| 9 | female | 22 | 162 | 22.0 | 2236 | 2080 | 2076 | 2258 | 39.36±4.45 | 0.91±0.22 |
| 10 | male | 32 | 189 | 32.1 | 2964 | 2522 | 4108 | 2832 | 42.40±4.19 | 0.87±0.17 |
| 11 | male | 24 | 188 | N/A | 1872 | 1820 | 1788 | 1942 | 39.62±4.47 | 1.10±0.31 |
| 12 | male | 24 | 180 | 19.6 | 2262 | 2310 | 1528 | 2304 | 40.83±4.27 | 0.93±0.27 |

## 1.11 Ablation Analysis

### 1.11.1 Effect of Training Snapshot Size

Figure 2 shows the single-radar classification accuracy (± standard deviation) for each task with respect to different training snapshot window sizes. As it is expected, increasing the training window size from one frame time (0.04s) to 1.12s leads to an increase in accuracy while resulting in a slower prediction.

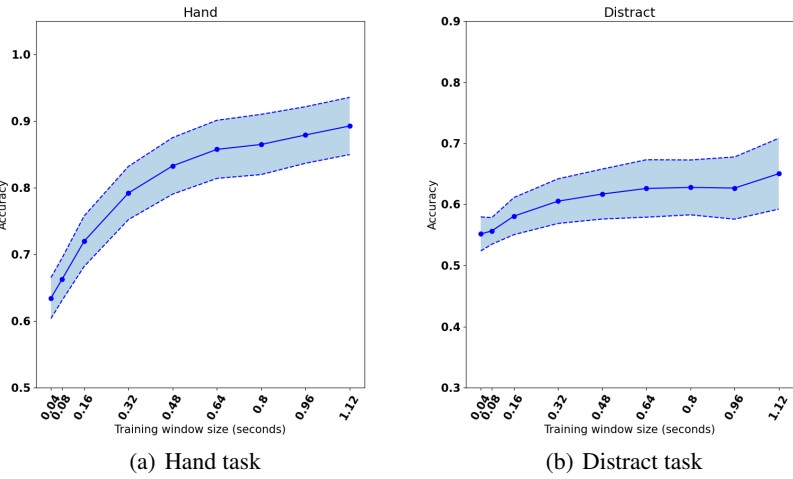

(a) Hand task      (b) Distract task

Figure 2: Single-radar classification accuracy across different training snapshot window sizes.

### 1.11.2 Effect of Baseline Model Size

Table 3 shows the single-radar classification accuracies ($\pm$ standard deviation) for the two tasks using different Convolutional Neural Nets as baselines. Apart from MobileNet, we used ResNet34 and EfficientNet B0 as two larger alternative baselines. However, no significant difference in accuracy was observed across the bench-marked baseline neural networks.

Table 3: Single-radar classification accuracy across different baseline models.

| Model | MobileNet | ResNet34 | EfficientNet B0 |
|---|---|---|---|
| Hand | 85.2±3.5 | 84.9±3.3 | 83.4±3.5 |
| Distract | 61.9±4.3 | 62.3±3.9 | 61.9±3.9 |