# OpenReview forum: "MVDoppler: Unleashing the Power of Multi-View Doppler for MicroMotion-based Gait Classification"
_NeurIPS.cc/2023/Track/Datasets_and_Benchmarks — NeurIPS 2023 Datasets and Benchmarks Poster_

### Official Review · Reviewer_EN5b · 2023-07-20
**Review for Submission 715**

**Rating:** 6
**Confidence:** 4
**Clarity:** The paper is well written.

**Strengths:**

The dataset seems to have properties not covered by existing data.  The authors have presented the paper in a clear manner with good figures.  The data was available on Google drive and seems well organized.

**Additional Feedback:**

None

**Correctness:**

The manuscript appeared correct and given the limitations of the dataset, the experiments were appropriate.  It would have been nice to see more experiments that examine things like how a model trained in one environment with distractor motions (e.g., cars) generalizes to another, but I realize that is out of scope of the dataset.

**Documentation:**

The dataset is a little light on documentation.  The data seem fairly simple to interpret, so this is not too much of a problem.

**Ethics:**

See comments on societal impact.

**Limitations:**

The authors did not do a very thorough job discussing the societal impact, particularly because the paper presents a method that could be leveraged in the context of surveillance.  It would have been nice to see a little more discussion on this.

**Opportunities For Improvement:**

The dataset is relatively larger than the other existing work. However, I didn’t do an exhaustive search of other gait datasets with micro-motions.  Although the dataset is larger than others it still is not very large, did I understand correctly that is contains a few hours (20,000+ seconds) of data?

It was difficult to find details in the paper about the subjects who are in the dataset.  While probably not as sensitive as computer vision tasks to appearance (e.g., height, weight, BMI etc.). I would have thought that in this dataset diversity of the people would have been pretty important.  How do the authors argue that their subjects are representative of a population (and which population)?

The motivation of the paper was a bit weak in my opinion.  The authors write: “Unlike camera-based systems that primarily capture an object’s shape, location, and color information (such as the shape and pose of a human body, the color of a car, or the dimensions of a helicopter), Doppler-based systems detect 39 the motions exhibited by subjects and their components (such as the motion of human hands or legs 40 during walking, the movements of a car’s wheels, or the rotation of a helicopter’s rotors).”  Is this really true?  Videos are used all the time to assess motion?  They are also very useful for providing contextual (e.g., visual) information about which motion are important.  I didn’t find it particularly convincing that a Doppler system would have advantages over computer vision, and importantly how the presented dataset is appropriate given that motivation.  I’d like the authors to add clarification there.

The baseline performance on some of the tasks seems fairly high (80-90% accuracy).  Can the authors comment on how much room for improvement they expect?  Given that the data were collected in a relatively constrained and contrived setting with a small number of people, I wonder if it is challenging enough or might benefit from greater diversity.

In table 1 the authors have a classification latency column - is classification latency a function of the dataset or the model/algorithm?  It seems a little confusing to me.

Overall, the paper is easy to read and relatively well presented, but it does seems light on some important details and motivation.

**Relation To Prior Work:**

The paper appears to have reasonable references to prior work.

**Summary And Contributions:**

This paper presents a dataset for gait classification based on Doppler radar signals.  The dataset capture “micro-motions” while people are walking.  The dataset contains labels for several classes and appropriate meta data.

---

> ### Author Response · Authors · 2023-08-21
> **Response to Reviewer EN5b 1/3**
>
> We want to thank the reviewer for their detailed comments and recognizing the novelty of the dataset and its advantages over previous state of the art.
> Below we have provided responses to each of the reviewer’s concerns:
>
> **[Q1] Dataset size** Although the dataset is larger than others it still is not very large, did I understand correctly that is contains a few hours (20,000+ seconds) of data?
>
> **[R1]** We have added the exact dataset size to the footnote of Table1 for further clarification. After data curation and cleap-up stages, MVDoppler contains more than 20 radar-hours of data (37,000+ seconds of recording per radar). We believe that with over 54,000 multi-view snapshots (diversified over different locations and trajectories), the dataset size is adequate for the defined tasks and the settings described in this paper. As evidence we have presented classification confidence bounds in Figure3 and histogram plots (with 30 to 100 snapshots per bin) in Figure2.
>
> **[Q2] Subject information** It was difficult to find details in the paper about the subjects who are in the dataset. While probably not as sensitive as computer vision tasks to appearance (e.g., height, weight, BMI etc.). I would have thought that in this dataset diversity of the people would have been pretty important. How do the authors argue that their subjects are representative of a population (and which population)?
>
> **[R2]** Table 2 in the supplementary materials includes information about each subject including their age, sex and height, as well as the average and standard deviation of their walking speed and SNR (as a proxy for Radar Cross Section). Our initial study did not consider recording BMI/Weight of participants. However, following the reviewers feedback we have contacted each participant regarding their weight/BMI and will include this information in Table2 conditioned on their cooperation and timely response.
> The population used in this study consists of volunteers (mostly grad-students) representative of healthy adults between the ages of 21 and 34. From the radar signal point of view, the factors most important in our dataset are the subject’s natural walking speed (variable between participants and also affected by factors like height) and their Radar Cross Section (also dependent on height as well as many other factors). The average height of participants in our dataset is 179cm for men and 165cm for women which compared with the US national average (175.3cm and 161.3cm correspondingly) is slightly higher, putting the population of our participants in the 67 percentile for both adult males and females in the US [1].
>
> [1] https://www.cdc.gov/nchs/data/series/sr_03/sr03-046-508.pdf

---

> > ### Author Response · Authors · 2023-08-21
> > **Response to Reviewer EN5b 2/3**
> >
> > **[Q3] Motivation** The motivation of the paper was a bit weak in my opinion... Videos are used all the time to assess motion? ... I didn’t find it particularly convincing that a Doppler system would have advantages over computer vision, and importantly how the presented dataset is appropriate given that motivation. I’d like the authors to add clarification there.
> >
> > **[R3]** There is no doubt that cameras (and other spatial sensors) are extremely powerful, and very useful in capturing positional context which can be useful for better analysis of the subject’s motion. The case made in this paper is not to replace cameras for gait classification tasks. Instead, we aim to demonstrate that an alternative and potentially complementary perception approach, leveraging completely different physics from typical optical sensors, exists that can succeed in some scenarios that cameras might fail. One example given in this paper is in the case of long-range perception (with high resolution demands on vision-based systems) but one could think of other scenarios like bad lighting or the effect of smoke and fog for use cases of an alternative non-optical perception approach.
> > It is true that cameras can be (and are) used for indirect measurement of velocity of objects (essentially by comparing the position of objects between two frames). However, there are three arguments that highlight the power of direct motion-analysis sensors (like radar) comparing to vision-based methods:
> > 1. Challenges in capturing fast movements:
> > Issues like motion blur and the rolling shutter effect are well-known issues that cameras face when trying to capture fast moving objects [2]. To provide a practical example, the radars used to capture the MVDoppler dataset measure movements of the subject every 299 microseconds (chirp rate). This translates to 3,344 scans of the subject every second. Cameras capable of operating at an equivalent frame rate do exist but they are out of the realm of inexpensive off-the-shelf cameras typically used for activity recognition applications. Moreover, motion estimation using high-speed cameras (using feature tracking or optical flow algorithms) is very computationally expensive [3]. We argue that although in principle it is possible to achieve high velocity measurement using high-speed global-shutter camera systems, using an off-the-shelf, inexpensive, direct velocity measurement sensor like mmWave radars is more practical.
> >
> > 2. Challenges in capturing small movements
> > Intuitively, the smallest motion measurable by a camera-based system is determined by its pixel size (in addition to  its frame rate). This means that even for very high speed cameras the velocity resolution of the camera is inversely proportional to the distance to the subject of interest. Additionally, calculating speed with cameras requires first detecting reliable “anchor points” between consecutive frames. This task is known to be increasingly difficult in lower-resolution, highly sensitive to the subject texture and lighting conditions, and computationally expensive [3,4].
> > As a practical example, consider that using a typical camera used in cars today (e.g., 1.2MP cameras used in Tesla Model 3 [5]) for analyzing hand movements of a pedestrian 30 meters away, requires reliable detection and tracking of anchor points described using only **1-3 pixels** and across multiple frames (assuming a uniform resolution across a 180 degree FOV and average human hand size of 13-25cm).
> > In contrast, Radar systems can **directly** measure micro-motions of subjects with high-precision, independent of their distance to the subject as long as we have sufficient SNR, and with computational overhead which is negligible compared to cameras.
> >
> > 3. Cameras and Radars can be complementary in motion analysis
> > As mentioned in the paper, Radar Doppler measurements are most precise in the “radial” direction (towards or away from the sensor). This is complementary to camera-based motion measurements which are most accurate when the subject moves across the pixels of an image (in the cross-range direction). Therefore combining radars with high-speed cameras can be a path to form the ultimate 3D sensor for micro-motion-based perception.
> > We hope that the explanation above helps convince the reviewer about the potential of Doppler-based systems as a less investigated perception solution that can be complementary, or alternative, to vision-based perception in specific applications.
> >
> > [2] Guo, Qing, et al. "Exploring the effects of blur and deblurring to visual object tracking." IEEE Transactions on Image Processing 30 (2021): 1812-1824.
> >
> > [3] Shah, S.T.H., Xuezhi, X. Traditional and modern strategies for optical flow: an investigation. SN Appl. Sci. 3, 289 (2021).
> >
> > [4] Kim, Yeon-Ho, Aleix M. Martinez, and Avi C. Kak. "Robust motion estimation under varying illumination." Image and Vision Computing 23.4 (2005): 365-375.
> >
> > [5] https://www.onsemi.com/products/sensors/image-sensors/ar0136at

---

> > > ### Author Response · Authors · 2023-08-21
> > > **Response to Reviewer EN5b 3/3**
> > >
> > > **[Q4] Room for improvement** The baseline performance on some of the tasks seems fairly high (80-90% accuracy).
> > > Can the authors comment on how much room for improvement they expect? Given that the data were collected in a relatively constrained and contrived setting with a small number of people, I wonder if it is challenging enough or might benefit from greater diversity.
> > >
> > > **[R4]** The reviewer has referred to the hand-detection task (with best single-radar accuracy of 86%) and have questioned the potential for accuracy improvements on this task and on the dataset as a whole.
> > >
> > > The hand-detection task is intentionally designed in a way that it can be solved virtually perfectly for the easiest scenario (e.g. 0-degree walking patterns). This is so that it can be used as a reference to measure the effect of walking location and trajectory on a “perfectly solvable task” and demonstrate that even in the worst case walking scenario (walking completely perpendicular to the radar’s beam axis), it is possible to have a highly accurate, while not perfect, classification. This baseline is designed to motivate researchers to close the gap between the worst case and best case scenario accuracy of the models through methods like the multi-view network fusion presented in the paper. The “Distract” task on the other hand is designed to be a more difficult counterpart to the “Hand” task providing researchers with a more difficult classification problem (single-radar baseline accuracy of 61-63%).
> > >
> > > To answer the reviewer’s question more quantitatively (what would be the maximum accuracy achievable for the hand-classification task?), we compare the accuracies reported in Figure 3 and take the accuracy achieved at the largest window size (94%) as a proxy for the best accuracy potentially achievable for the real-time hand-movement detection task with a snapshot size of 640ms.
> > > We agree with the reviewer regarding the comment that adding more diversity to the dataset (in terms of number and diversity of participants, number of classes etc) will help in defining more challenging and interesting tasks in future work. We hope that MVDoppler will be seen as a significant first step in motivating capture and publication of carefully curated as well as diverse datasets that will be beneficial to both Radar and Machine Learning communities.
> > >
> > > **[Q5] Latency** In table 1 the authors have a classification latency column - is classification latency a function of the dataset or the model/algorithm? It seems a little confusing to me.
> > >
> > > **[R5]** We recognize that usage of the term latency might be confusing and have added a footnote to Table 1 to clarify it better.
> > > The latency mentioned in this table is maximum end-to-end classification latency acceptable for each specific task. In gait analysis applications the analyzed phenomena (human walking cycle) is usually much slower than typical neural net inference times (around 0.5-1 seconds per step vs tens of milliseconds per inference). Therefore the latency reported in Table1 mostly refers to the minimum observation window length required from the model to perform an specific task. In typical Doppler datasets, the observation time can be in order of 5-15 seconds (10 to 30 steps). In MVDoppler, we assumed an observation time equivalent to a single human step (640 ms) aiming to show potential of Doppler-based methods for close-to-real-time applications.
> > >
> > > **[Q6] Societal Impact** The authors did not do a very thorough job discussing the societal impact, particularly because the paper presents a method that could be leveraged in the context of surveillance. It would have been nice to see a little more discussion on this.
> > >
> > > **[R6]** Thanks for your comment. Please see the added details in the societal impact section (Section A.1 in Supplementary materials).

---

> > > > ### Comment · Reviewer_EN5b · 2023-08-29
> > > > **Response to Authors**
> > > >
> > > > The author's response is thorough and I appreciate the extra details and the effort to recontact participants for additional information about their weight/BMI.  I also appreciated the added information about the broader impact of the work.  I am happy to increase my score and recommend that this paper is accepted.

---

### Official Review · Reviewer_MPVr · 2023-07-21
**MVDoppler is a good multi-view Doppler dataset for learning from distributed micro-motion radars**

**Rating:** 7
**Confidence:** 5

**Strengths:**

- clear, thorough, and well-argued discourse

- rigorous experimental design and dataset statistics

- good variants of models to establish a 1st baseline on multi-view Doppler learning


**Additional Feedback:**

2) Related work

    - 2.1:

        * Clear summary and differentiation from related work

    - 2.2

        * Wording of lines 113-114 a bit confusing
        * References are adequate, but some favour original works, while others favour certain technical communities
        * Good summary claim at end of section. I wonder if this lays foundations to ambitious "generative" modelling of radar signals

3) Dataset

    - 3.4:

        * Rigorous experimental design and dataset statistics
        * It would be good to comment on how these statistics were generated, perhaps in supplementary
        * Fig. 2c, good statement

4) Single-radar baseline

    - 4.2:

        * Std range in Fig. 3 is perhaps better shown as a shading as is common practice in community
        * No implementation details in supplementary

5) Multi-radar baseline

    - 5.1:

        * Concatenated in the channel domain -> channel-wise
        * Lines 228-234 clunky, consider rewording (e.g., average -> combine additively?)
        * 5 multi-view benchmarks -> variants. Use of term benchmark a bit confusing as it often refers a dataset + set of techniques rather than variants of a method.

6) Results

    - Lines 253-256: Very well designed and controlled experiments highlight foundational properties of Doppler-based sensing
    - Lines 258-266: Ditto
    - It would be good to provide an appendix to review the dependence of Doppler on angle for a self-contained treatment

**Clarity:**

- Clear, thorough, and well-argued work

**Correctness:**

- claims are well supported

- methods and experiments are good

**Documentation:**

- authors provide supplementary material and code repo to document dataset and baselines

- a dataset snapshot (~25GB) is provided, but this reviewer didn't check beyond size

**Ethics:**

Authors comment on any relevant ethical concerns in supplementary material.

**Limitations:**

As pointed out by the authors (lines 268-271), the dataset is curated under somewhat idealised conditions (e.g., absence of aggressive multipath and clutter). Additionally, the authors remarked on the potential to push the envelope of micro-Doppler analysis for more challenging perception task (lines 275-279); for example, one such very challenging task to to fully identify a subject from their gait signature. Nonetheless, this reviewer believes that the authors have pretty much laid bare the facts for future work.

**Opportunities For Improvement:**

- relatively limited bandwidth due to frequency multiplexing

- it would be good to provide an appendix to review the dependence of Doppler on angle (and dwell time) for a self-contained treatment for this more ML-minded community

**Relation To Prior Work:**

References are adequate, although some favour original older works, while others favour certain technical communities.

**Summary And Contributions:**

This paper introduces a large-scale Doppler dataset that supports multi-view (i.e., multi-device) learning. The authors also implement a suite of perception models to go with the dataset in order to establish a baseline against which future research can be evaluated. Thorough and carefully designed analysis provides a summary of expectations on learning from multi-view Doppler radars and as such lays the foundations for future research.

---

> ### Author Response · Authors · 2023-08-21
> **Response to Reviewer MPVr**
>
> We thank the reviewer for their detailed feedback and their recognition of the value and potential of MVDoppler as the first public multi-view Doppler dataset for gait analysis applications.
>
> Below are our responses to the reviewer’s comments and suggestions.
>
> **[Q1]** Wording of lines 113-114 a bit confusing.
>
> **[R1]** Thanks for the suggestion. We have revised the corresponding sentences.
>
> **[Q2]** I wonder if this lays foundations to ambitious "generative" modelling of radar signals.
>
> **[R2]** Thanks for your comment and interesting idea about the generative modeling of radar signals. We agree that working towards generative modeling of radar signals is definitely a very interesting research path with applications both in design of radar simulators and in forming robust embedding spaces for radar signal compression and fusion. This will only be possible through a common effort by the radar community to capture and share radar datasets at a massive scale. Our hope is that MVDoppler motivates capture and public release of similar datasets.
>
> **[Q3]** It would be good to comment on how these statistics were generated, perhaps in supplementary.
>
> **[R3]** Thanks for your comment. Related section added in the supplementary materials.(Section A.7)
>
> **[Q4]** Std range in Fig. 3 is perhaps better shown as a shading as is common practice in community.
>
> **[R4]** Thanks for your comment. Figure updated as suggested.
>
> **[Q5]** No implementation details in supplementary
>
> **[R5]** Thanks for your comment. Implementation details added in the supplementary materials.(Section A.7)
>
> **[Q6] Typos and wording in section 5.1** Concatenated in the channel domain -> channel-wise. Lines 228-234 clunky, consider rewording (e.g., average -> combine additively?). 5 multi-view benchmarks -> variants. Use of term benchmark a bit confusing as it often refers a dataset + set of techniques rather than variants of a method.
>
> **[R6]** Thanks for your comment. We have updated the wording as suggested.
>
> **[Q7]** It would be good to provide an appendix to review the dependence of Doppler on angle for a self-contained treatment
>
> **[R7]** Thanks for your comment. Related section (Section A.8) added in the supplementary materials.

---

> > ### Comment · Reviewer_MPVr · 2023-08-28
> >
> > Dear authors,
> >
> > Many thanks for acting on feedback with improved manuscript and supplementary. This is a valuable work that goes a long way towards advancing distributed multi-radar fusion research, which should be accepted.

---

### Official Review · Reviewer_reAu · 2023-07-21
**Review for MVDoppler: Unleashing the Power of Multi-View Doppler for MicroMotion-based Gait Classification**

**Rating:** 6
**Confidence:** 2
**Correctness:** The dataset appears sensible and the …

**Strengths:**

1. Compared to existing doppler datasets as described by the authors, their dataset has a better variety of micro-doppler across both range, azimuth, and motion direction. It is also longer, though includes a slightly more limited set of participants than other datasets. It additionally includes motion from hands that occurs while the participant is walking rather than standing.

2. As a promising and under-explored area of perception, this dataset is potentially valuable in enabling the development of more radar/doppler based systems.

3. The paper includes a valuable analysis both of the distribution of data contained (in the form of Figure 2's heatmaps) and the abilities/failure modes of baseline methods on the data (Figure 5).

**Additional Feedback:**

N/A

**Clarity:**

The paper is presented clearly and simply. Figures are easy to understand and immediately inform the reader of the dataset setup and design.

**Documentation:**

The code and dataset toolbox is already released, which is very promising. The supplement contains important technical details regarding how the doppler data was collected. The data is already online and a data maintenance plan associated with a university is included.

**Ethics:**

Human participants faces are blurred in the data but it is unclear whether they may be recovered somehow. There don't appear to be other ethical concerns.

**Limitations:**

1. Although clearly a promising extension on existing doppler-based datasets, this dataset is still from a relatively contrived setup that it's not clear will lead to significant improvements in methodology.

- For example, the radars remain static, though clear use-cases for such systems are from moving radars that are mounted on cars. Given doppler is a signal that involves understanding how objects have changed over time, moving radars on a track might be more directly useful in developing such radar-perception systems.

- A different fusion system that is considerate of real-world applications might have placed two radar setups in stereo with one another, as if they were setup a few feet apart on a car. Such stereo-fusion would be another interesting analysis (and potentially applicable).

2. Coming from only a few hours of recording, the dataset feels more like a proof-of-concept of an improved doppler dataset, and could benefit from more involved actions and benchmarks. Specifically, good datasets are often forward-looking, even if they have insufficient baselines. For example:

- Even if baselines struggle at these tasks, one could have included a few dozen more actions including gaits (running, skipping, biking).

- Even if there's no consideration of analyzing this, one could have recorded multiple people crossing paths with one another. Such a complicated environment is not necessary for nascent first-of-its-kind-contributions, but could be useful for further, more developed methods. Having this data can be what really makes a dataset useful (even when it's unclear how any method might use it). Once the recording setup is established, it appears like it shouldn't be too difficult to record such a diversity and produce a better, more expansive dataset.

- More distractor setups could exist, for example someone walking while reading a book or walking while looking away from the direction of the radars. This last one might be particularly difficult but potentially closer to the essence of what it means to be "distracted".

**Opportunities For Improvement:**

The paper is high-quality and has done thorough analysis/consideration along the axes it emphasizes. See limitations below.

**Relation To Prior Work:**

Prior doppler datasets are briefly mentioned, with a helpful table summarizing their drawbacks and limitations. It is shown how MVDoppler improves upon a number of drawbacks of prior works.

**Summary And Contributions:**

MVDoppler is a dataset of recorded radar (and maybe RGB) from two radars mounted around a singular RoI roughly 10 meters square, setup on an open grassy field. The radars are setup at 90 degrees to one another, with each facing a side of the square. It contains 10-second episodes of people walking within this square, and these are divided into short clips from which classification may be done. Overall, there are a few hours (>20,000s) of doppler readings, with 10-20 participants total.

Two benchmark tasks, hand classification and distraction are introduced. Hand classification consists of determining whether hands are dynamic and moving (or static as in pockets or using a phone). Distraction consists of determining whether a person is on a phone call or texting. Participants enter the RoI and either walk along snake-like-s patterns or spirals or "randomly".

The overall goal is to contribute data that enables new methodologies that can use doppler radar to understand pedestrians/people in scenes, particularly their gait and distraction. This is a potentially valuable direction for self-driving systems and other autonomous systems that operate in the built environment.

---

> ### Author Response · Authors · 2023-08-21
> **Response to Reviewer reAu**
>
> We want to thank the reviewer for recognizing the value of the published dataset and their thoughtful feedback on the different ways to improve the dataset. Although we did not find a specific question from the reviewer, we have provided clarifications in response to the reviewer's concerns on the limitations of the presented work below.
>
> **[Q1] Relevance of large baselines**
>
> **[R1]** As the first public multi-view dataset for micro-motion-based perception, the motivation of our work goes beyond only autonomous driving applications. The concept of micro-motion-based perception can have several other applications in security, retail, gaming and entertainment, in which static large-baseline setups like the one used to capture MVDoppler are common. Moreover, our baselines include both single-radar classification (suitable for autonomous driving applications) as well as multi-radar experiments (suitable for large-baseline static setups).
>
> **[Q2] Effect of radar motion**
>
> **[R2]** Regarding platform motion, it should be noted that even in case of autonomous driving scenarios, one would first compensate for the effects of radar movements on the Doppler signal before it can be used for gait classification. Therefore the problem of radar motion compensation is usually studied separately from the gait classification task. To get a sense about the effect of unpredictable radar platform motions (like engine vibration) on classification accuracy of the specific tasks defined in MVDoppler the readers are referred to a recent paper on the subject [1].
>
> **[Q3] Limited dataset size**
>
> **[R3]** We want to bring to the reviewer’s attention that the tasks defined in this work rely on classifying micro-motion patterns in a time scale as short as **a single human step**. Therefore, although with around 20 radar-hours of data, MVDoppler is relatively small compared to common video datasets, it still includes over 108,000 of single-radar snapshots that we believe can sufficiently capture the effect of walking trajectory and location on each task’s accuracy.
>
> **[Q4] Multi-subject experiments**
>
> **[R4]** Similar to the case for platform motion, a common assumption in gait analysis literature is that the Doppler signature of the subject of interest has already been extracted by a detection and tracking algorithm prior to gait classification. Moreover, we believe that eliminating the need for detection and tracking through including only one subject in the RoI, helps with removing dependency of the gait classification results on a specific tracking algorithm’s success and failure patterns.
> As another benefit of controlled experiment design, it should be noted that the published dataset can easily be used to simulate multi-person scenarios (with a controllable diversity of walking patterns and the number of people simultaneously walking) by simply combining the raw time series data recorded from different single-subject experiments.
>
> **Finally,** we want to express appreciation for the reviewers detailed comments, suggestions, and ideas which help us to improve the future versions of the dataset specially in terms of class diversity and experiment design.
>
> [1] S. Hor, N. Poole and A. Arbabian, "Single-Snapshot Pedestrian Gait Recognition at the Edge: A Deep Learning Approach to High-Resolution mmWave Sensing," 2022 IEEE Radar Conference (RadarConf22), New York City, NY, USA, 2022, pp. 1-6, doi: 10.1109/RadarConf2248738.2022.9764196.

---

> > ### Comment · Reviewer_reAu · 2023-08-26
> >
> > I especially appreciate the authors highlighting how their proposed dataset is applicable to micro-motion patterns that operate on a very small timescale. This clarification is valuable for positioning the work within a broader context of computer vision tasks and data.

---

### Official Review · Reviewer_cCoP · 2023-08-04
**Review for MVDoppler**

**Rating:** 7
**Confidence:** 3
**Correctness:** The results and analysis are done wit…
**Clarity:** Clear work and presentation as well a…

**Strengths:**

New dataset for using doppler for looking into gait

**Additional Feedback:**

Good for publication

**Documentation:**

The datasets are presented nicely.

**Ethics:**

Ethics approval is present.

**Limitations:**

As mentioned this is highly susceptible to test conditions and the presence of multi subjects or different types of movements can impact the accuracy of the work.

**Opportunities For Improvement:**

The work can be very limited to the conditions of experimentation. Slight changes to that can change the results. Sensitivity to environment would be great to be tested.

**Relation To Prior Work:**

The authors have discussed the prior work in detail.

**Summary And Contributions:**

The authors present a dataset using doppler sensor for glassification of gait. This is an interesting work and new approach.

---

> ### Author Response · Authors · 2023-08-21
> **Response to Reviewer cCoP**
>
> We want to thank the reviewer for recognizing the value and potential of our work. Including a sensitivity analysis to environmental factors is one of our main goals in future work.

---

### Decision · Program_Chairs · 2023-09-22

**Decision:**

Accept (Poster)

**Comment:**

This paper presents a dataset of multi-view doppler data for gait classification.  The reviewers all agree that this dataset has merits and all reviewers recommended that this paper was above the acceptance threshold.  The strengths of the paper were: 1) that it contributes to an underrepresented domain, that the paper was clear and well presented, and that the dataset has properties that are not covered in other datasets. The setup of the data collection is a little contrived and the scale is a little small; however, on balance the reviewers agree that this dataset is a valuable resource for the community.